# Transvaginal Strain Elastosonography May Help in the Differential Diagnosis of Endometriosis?

**DOI:** 10.3390/diagnostics11010100

**Published:** 2021-01-09

**Authors:** Gábor Szabó, István Madár, Attila Bokor, János Rigó

**Affiliations:** 1Department of Obstetrics and Gynecology, Faculty of Medicine, Semmelweis University, Baross utca 27, 1088 Budapest, Hungary; madar.istvan@med.semmelweis-univ.hu (I.M.); bokor.attila@med.semmelweis-univ.hu (A.B.); rigo.janos@med.semmelweis-univ.hu (J.R.J.); 2Department of Clinical Studies in Obstetrics and Gynecology, Faculty of Health Sciences, Semmelweis University, Vas utca 17, 1088 Budapest, Hungary


**Dear Editor,**


We read with great interest the paper entitled “Differential Diagnosis of Endometriosis by Ultrasound: A Rising Challenge” of Scioscia et al. [1]. We fully agree with the authors that with the spread of the International Deep Endometriosis Analysis (IDEA) group protocol, the detection rate of endometriosis has significantly increased among users. As the number of detected cases increases, so does the number of false positive results. Because of the heterogeneous, multiform and often non-specific symptoms of endometriosis, differential diagnostic is a real challenge. The authors’ excellent article provides a comprehensive review of the possible involved organs.

They correctly pointed out that deep endometriosis infiltrates the anterior wall of the rectum. Endometriotic nodules affecting the rectum and rectosigmoid bowel are hypo- or anechogenic. Due to scarring, they show with color Doppler only a minimal signalling. We have previously reported that infiltrating lesions of rectal endometriosis appear to be stiffer than sections of healthy intestinal wall by transvaginal strain elastosonographic examination [2] (Figure 1A). A strain ratio (SR) of 2.0 serves as a cut-off value for the optimal distinction.

However, stenosis of the rectosigmoid bowel is not only caused by endometriosis. Colorectal cancer has the highest incidence rates of all gastrointestinal malignancies worldwide and also leads to changes in the ultrasound image of the bowel [3]. Rectal cancer usually grows outward and it can reach the serosa, contrary to deep infiltrating endometriosis lesions infiltrating inward. With a color Doppler ultrasound examination, rectal cancer shows increased vascularity [4]. Previously with transrectal elastosonographic examination, benign adenomas did not show a significant difference compared to a healthy intestinal wall (SR ≤ 1.25). Malignant tumors have an SR > 1.25 [5]. Stiffness of early rectal cancer—T1 and T2 stages according to the TNM classification of malignant tumors—in contrast to deep endometriosis, is no more than twice that of a healthy, intact bowel wall [6]. There is a tendency for a higher fibrosis score (SR > 2.0) only for tumors staged as T3 or T4. This observation may provide additional information for differential diagnosis. In our practice, a 37-year-old patient with suspicious signs for deep endometriosis (infertility, diarrhoea and bloating) was examined with transvaginal sonography. On the anterior wall of the rectum, a stenosing lesion of 2 cm in diameter was visible at a 15 cm distance from the anal verge. Color Doppler showed intense vascularisation. The layers of muscularis propria and subserosa seemed to be intact. With strain elastography, the serosa was without interruption. The strain ratio between lesion and normal bowel wall was 1.33 (Figure 1B). Histological examination confirmed well-differentiated adenocarcinoma from the sample, obtained during the performed colonoscopy.

Inflammatory bowel diseases (IBDs) are like endometriosis, also being common pathologic conditions of unknown origin. Epidemiological studies reported a positive association between endometriosis and IBD [7]. Crohn’s disease occurs predominantly in young women in their peak reproductive years. Inflammation of the intestinal wall can lead to adhesions, perforations and fistula formation. Later, it can also cause fibrosis in the bowel wall. No correlation was previously observed between mean strain ratio and fibrosis score [8]. In Crohn’s disease, similar to endometriosis, patients may develop symptoms like bloating, haematochezia and diarrhoea. In our practice, a 35-year-old patient with such complaints was diagnosed with deep infiltrating endometriosis in the rectovaginal space with palpation. Colonoscopic examination showed protrusion in the lumen of the rectum at a 15 cm distance from the anal verge. Transvaginal strain sonoelastography showed a stiff nodule of 3 cm in diameter. In the same patient, transvaginal ultrasound also revealed, on the right side, adhesions between the small intestine and thickening of the intestinal wall. A conventional ultrasound image of the intestinal conglomerate was similar to multicentric deep endometriosis in the bowel, but the elastostonography did not confirm this. Elastosonographic examination could not isolate a stiff nodule corresponding to deep endometriosis. Instead, a pattern suggestive of diffuse fibrosis appeared (Figure 1C). During surgery in the rectovaginal space, we excised a deep endometriosis nodule of 3 cm in diameter. Histological examination verified endometriosis. In the ileocoecal region, intraoperative peritoneal adhesions and bowel strictures were found resembling endometriosis, but histology confirmed Crohn’s disease only.

This is the first case, to the best of our knowledge, in the literature where preoperative differential diagnosis has been achieved with the use of elastosonography by the simultaneous occurrence of deep endometriosis affecting the gastrointestinal tract and Crohn’s disease.

As Scioscia et al. conceive in their article, “accurate diagnosis of endometriosis has significant clinical impact and is important for appropriate treatment”. To achieve this, an interdisciplinary approach and the appropriate application of different imaging methods are necessary. Transvaginal ultrasound is a widely available, non-invasive technique. In our opinion, sonoelastography can provide additional data as a valuable element of the diagnostic process. It is important for the physician performing the sonography to receive feedback from the surgeon or even be present at the surgery.

## Figures and Tables

**Figure 1 diagnostics-11-00100-f001:**
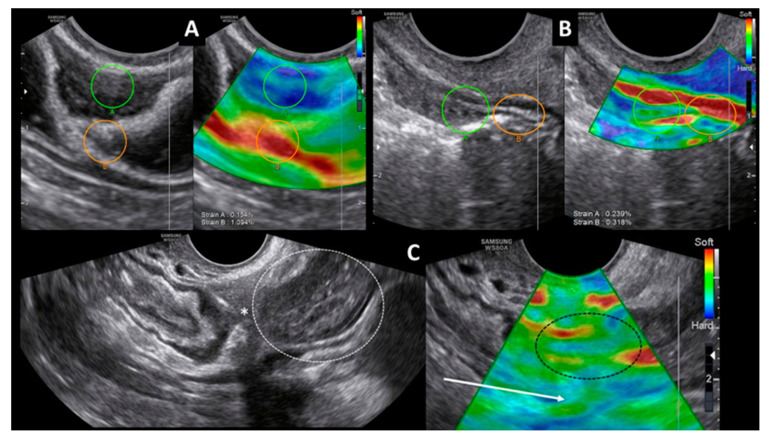
Transvaginal strain elastosonography in different intestinal diseases. Dual-mode greyscale sonographic (left) and strain elastosonography (right) images. (**A**) Deep infiltrating endometriosis on the anterior wall of the rectum. A stiff delineated lesion (blue area) infiltrates through the serosa into the muscularis propria layer. The posterior regular rectal wall is soft (red area). Strain measurement between the lesion in the anterior rectal wall (green circle) and the regular posterior rectal wall (orange circle). The calculated strain ratio (SR) is 7.1. (**B**) Adenocarcinoma on the anterior wall of the rectum. The sonoelastographic appearance of the tumor shows increased stiffness (blue and green area). The serosa of the rectum is uninterrupted (red area). Strain measurement between the tumor (green circle) and the regular rectal wall (orange circle). The calculated strain ratio (SR) is 1.33. (**C**) Sagittal transvaginal sonogram in Crohn’s disease. On the greyscale sonographic image (left), thickened intestinal wall (circle) and stricture (asterisk) are visible in the ileocoecal region. The sonoelastographic appearance (right) of the intestinal loops does not show a delineated nodule in the area of the stricture (black circle) but diffuse fibrosis between the adhering intestinal loops (arrow).

## Data Availability

No new data were created or analyzed in this study. Data sharing is not applicable to this article.

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
