# Peer review of "Transvaginal Strain Elastosonography May Help in the Differential Diagnosis of Endometriosis?"

_diagnostics, 2021, doi:10.3390/diagnostics11010100_

Round 1
Reviewer 1 Report
Authors proposed the comments for transvaginal strain elastosonography for endometriosis diagnois. They confirm their idea using dual-mode gray-scale sonographic and strain elastoongraphy images with suspicious deep endometriosis signs.Histological examination confirmed well-differentiated adenocarcinoma signs in Crohn's disease. Compared to conventional ultrasound image, transvaginal ultrasound could reavel the adhesions between the intestines and thickening of the intestinal wall.
This could be very interesting approach using elastosonography for deep endometriosis in Crohn's disease. There are no English grammar mistakes. Figures are very clear to be seen. Therefore, the comment could be accepted with some minor suggestion.
1. Authors need to provide the author contributions.
2. Authors need to use abbreviated journal titles in the Reference section.
Author Response
Dear Reviewer,
Thank you very much for reviewing our manuscript carefully. We respectfully accept your suggestions. We will respond to your comments point by point.
- Authors need to provide the author contributions.
Author contributions have been entered according to your instructions.
- Authors need to use abbreviated journal titles in the Reference section.
In the original draft of our manuscript, we used the abbreviated journal titles in the Reference section used by Pubmed. In the revised version we changed the abbreviations of the journal titles according to the Web of Science Journal Title Abbreviations by Clarivate.
Changes are marked in green in the manuscript.
Sincerely

Reviewer 2 Report
I reviewed the comment entitled “Transvaginal strain elastosonography may help in the differential diagnosis of endometriosis?” submitted by Szabo et al for consideration to Diagnostics. I found it an interesting paper of clinical importance with a meticulous description of an interesting case of simultaneous occurence of deep endometriosis of the gastrointestinal tract and Crohn's disease. From my point of view, this manuscript warrants publication with some minor changes that are listed below.
More specifically,
Page 2, lines 41-43 and lines 44-45: Please add references.
Author Response
Dear Reviewer,
Thank you very much for your thorough review of our manuscript. We respectfully accept your suggestions. We will respond to your comments point by point
- Page 2, lines 41-43 and lines 44-45: Please add references.
In accordance with the instructions, two references were added to the manuscript regarding the ultrasound characteristics of rectal cancer.
- Scioscia, M.; Orlandi, S.; Trivella, G.; Portuese, A.; Bettocchi, S.; Pontrelli, G.; Bocus, P.; Anna Virgilio, B. Sonographic Differential Diagnosis in Deep Infiltrating Endometriosis: The Bowel. BioMed Res Int. 2019, 1–9, doi:10.1155/2019/5958402.
- Sudakoff, G.S.; Quiroz, F.; Foley, W.D. Sonography of anorectal, rectal, and perirectal abnormalities. Am J Roentgenol. 2002, 179, 131-136.
Changes are marked in yellow in the manuscript.
Sincerely
